# Environmental Health Knowledge, Attitudes, and Practices of French Prenatal Professionals Working with a Socially Underprivileged Population: A Qualitative Study

**DOI:** 10.3390/ijerph16142544

**Published:** 2019-07-16

**Authors:** Marion Albouy-Llaty, Steeve Rouillon, Houria El Ouazzani, Sylvie Rabouan, Virginie Migeot

**Affiliations:** 1INSERM, University Hospital of Poitiers, University of Poitiers, Clinical Investigation Center (CIC) 1402, 86021 Poitiers CEDEX, France; 2Department of Public Health, BioSPharm Pole, University Hospital of Poitiers, 86021 Poitiers CEDEX, France; 3Dispositif de recherche Interventionnelle en promotion de la santé environnementale, Faculty of Medicine and Pharmacy, University of Poitiers, 860310 Poitiers CEDEX, France; 4CNRS UMR 7285, Institut de chimie des milieux et matériaux de Poitiers, IC2MP, University of Poitiers, 86073 Poitiers CEDEX, France

**Keywords:** environmental health, inequalities of health, emergent risks, pregnancy, health education, medical education

## Abstract

Introduction: As environmental health knowledge of population is associated with social economic status, the objective of this study was to determine environmental health knowledge, attitudes, and practices of French prenatal professionals working with a socially underprivileged population. Material and methods: A focus group with eleven prenatal professionals working with socially underprivileged population was carried out in France in 2015. Content analysis of verbatim explanation was conducted with choice-of-subject categories carried out according to the triangulation principle, and topic trees were generated and applied. Results: The professionals have non-specialized experimental knowledge of emergent risks and were essentially preoccupied by infectious biological risks. In practice, however, they became increasingly cognizant of emergent risks. Their educational practices take cultural context into account but educational tools with imagination and affectivity have to be developed in order to reach socially underprivileged population. Discussion: Professionals are often sensitized to the field of environmental health in their apprehension of polluted biological environments, which they associate with social contexts and behavioral factors. In this study, we recommend adapted training programs and updated professional guidelines in view of reducing prenatal emergent risk exposures and social inequalities of health.

## 1. Introduction

French prenatal professionals do not hesitate to deliver advice to all pregnant women on numerous subjects such as diet, physical activity, self-medication, and consumption of psychoactive substances (alcohol, cannabis, tobacco, etc.); occupational hazards, unfavorable socio-economic conditions, ill-treatment, domestic violence, and other factors of vulnerability that can be at the root of subsequent difficulties. Guidelines and recommendations for pregnant women are given in an official document by the French Authority of Health, la Haute Autorité de Santé (HAS) [1]. However, no mention is made of exposure to emergent risks such as endocrine disruptors, chemical substances present in numerous sectors, and daily objects such as plastics, cosmetics, and food, even though the medical community has begun to show interest in the problem because of the suspicion about their adverse impacts on the human health [2]. Indeed, as they are likely to affect the major functions of reproduction, growth and development, and homeostasis and metabolism, endocrine disruptors are suspected to increase occurrence of cancers, prematurity, fetal growth, obesity or degenerative diseases [3].

So, maternity hospitals now issue brochures on exposure to endocrine disruptors [4] and pregnant women are advised, for example, not to color their hair with chemical products and to reduce the use of cosmetics and lotions [5].

Contrary to tobacco exposure during pregnancy, which remains the principal risk referred to by perinatal health professionals [6], exposure to emergent risks is not mentioned in the existing French existing guidelines [1] but in American’s [7].

In fact, even though professionals are largely convinced that prenatal advice and recommendations can help to reduce exposure to environmental pollutants, few of them include this theme in their prenatal consultations. A number of obstacles have been observed, one of which is a lack of competence in environmental health [6].

Moreover, population knowledge about emergent risks is associated with social economic status [8] and pregnancy anxiety [9]. Women with better knowledge are more educated and anxious. For underprivileged populations, health education needs other methods than those for the general population. Imagination and affectivity have to be developed and cultural context has to be taken into account [10].

To date, among the studies on the knowledge, attitudes, beliefs, and practices of prenatal professionals in the field of environmental health [6], only 7 out of 43 are European. Three recent studies took place in France [11,12,13] but did not focus on vulnerable populations. We conducted a qualitative study whose objective was to identify the knowledge, attitudes, and practices on environmental health of French prenatal professionals working with an underprivileged population.

## 2. Materials and Methods

We carried out a focus group discussion session.

### 2.1. Participant Selection

The target population was composed of French prenatal health care professionals working with underprivileged population.

### 2.2. Sampling

All the professionals came from the district office of maternal and childhood protection (Protection Maternelle et Infantile, the French PMI): eight midwives, one nursery nurse, one general practitioner, and one obstetrician. PMI is a public health ward from the department council created in France in 1945. This ward has offices in medico-social centers and work with perinatal nurses, psychologists, doctors, midwives, young children’s educators, and the staffs of activity centers. PMI offers medical, psychological, social, and educational prevention measures for future parents and children who are 6 years old or below. For example, PMI offers screening of disabilities during childhood and visits during perinatal period to underprivileged families.

### 2.3. Methods of Approach

A training day in environmental health was organized as part of an interventional research study called “PREVED” (Pregnancy, Prevention, Endocrine Disruptors). The program is presented in Table 1.

The first training sequence used a photolanguage^®^ method. In this method, photos serve as a support for speech so that each one expresses his experiences or representations of a theme. Pictures are placed on a table around which participants can turn. The session begins with a question carefully prepared by the facilitator that leads to the choice of pictures to be silently observed. After the choice, everyone presents his or her photograph whenever he or she wishes, bouncing back or not with the previous speech; others listen carefully without comment. On the training day, the theme chosen was environmental health from a non-professional, “citizenly” standpoint. The expected centers of interest were rapidly and spontaneously alluded to: indoor air, hygiene products and cosmetics, the outdoor environment, electromagnetic fields, and food consumption. These different themes were generally considered in terms of solutions, not risks.

After that, a conference was given on environmental health and the limits of science in this field (S.Ra). The professionals were indeed aware of the boundaries of existing scientific research and of the need to promote health education with these limitations in mind [14].

The third step was dedicated to professional attitudes with regard to environmental health education. A picture illustrating the visit of a PMI professional in a child’s room at home was proposed for examination. Professionals were asked to comment on it, and their commentaries were at the heart of the focus group proceedings presented in this paper. The accompanying grid was built around two basic questions: Looking at this picture, what are the environmental health issues you discuss with families? How do you proceed?

### 2.4. Sample Size

Eleven persons participated. The professionals (midwife, perinatal nurses, gynecologist, general practitioner) all knew each other before the focus group was formed. As the department head was present, there existed a hierarchical relation between the participants, all of whom were French women. The median age was 42.

### 2.5. Non-Participation

There were no non-participants.

### 2.6. Data Collection

The focus group session lasted 62 minutes. It was recorded in the presence of an introducer (C.M.) and an observer (M.A.L.) tasked with noting the non-verbal language of each participant. The focus group session ended when the ideas had reached saturation and when no new information was being given, in accordance with focus group methodology.

### 2.7. Setting

The focus group session took place in September 2015 in a health environment education setting, that is to say an apartment (“Atelier du 19,” in Poitiers, France) commonly used to further pedagogical initiatives in environmental health.

### 2.8. Analysis

It was carried out by content analysis of verbatim explanation. It consisted in three phases: extraction of all information, detection of the relevant data and organization of topic trees. Some of the themes had been preliminarily identified in a previous qualitative study [9].

We applied the triangulation method of analysis. The verbatim was individually read by four members of our team and each extracted idea was coded. They then met for an exchange of ideas designed to identify and to reach a consensus on the key themes. The ideas had to be of an exclusive nature, formulated without personal interpretation and correctly associated with one another, within the framework of the key themes if necessary.

The data were sorted and ordered using the RQDA qualitative analysis software CAQDAS^®^ (Computer-Assisted Qualitative Data Analysis Software) (University of Surrey, UK) and implementing the [R]^®^ (R development core team, R Foundation for Statistical Computingc/o Institute for Statistics and Mathematics, Vienna, Austria) program.

This study has been approved by person protection committee with international review board approval (IRB) approval number: 2015-A00031-48.

## 3. Results

All the verbatim coming from the participants are presented in italics.

### 3.1. Perception of Knowledge

It seemed to the observer that the environmental health field was more easily known by the participating health care professionals during the first training sequence, when they were asked to express themselves as concerned citizens. During the third sequence, their remarks on prevention were essentially based not only on HAS guidelines, but also, at times, on their non-specialized knowledge.

Prior to the focus group activity, all eleven participants stated that they did not feel legitimately entitled to speak out on the relationship between health and environment. They readily recognized the limitations of their knowledge: “*I do not know from what standpoint the subject can be approached.” “Pesticides, they surely create issues like so many other problems we do not even suspect.” “We do not necessarily have the answers!”* “*As we have not received all the messages, we do not know where we wish to head.” “With other problems, we feel, how can I put it, less safe?” “And so it is that it doesn’t stick quite as well? Water-based paint is less cohesive than paint, I don’t know, with solvents?” “And so, as it is known, what I know!”*

Non-specialized knowledge permeated remarks such as “It is known, I know.” “We feel there are matters with which you are at ease.” “What belongs to personal knowledge and to common knowledge, that’s what it is!” “Topics on which we are going to be more centered.” “Each of us has her own little thing, her own hobbyhorse.” “Yes, but what I once heard on the radio was that to remove all traces of pesticides from a fruit, it needs to be peeled centimeter by centimeter...”

However, the professionals frequently referred to the HAS guidelines, which they scrupulously respect: “I spoke of what is written in the texts on preparation for childbirth, the guidelines, and the subjects which we have got to approach during the preparation.” “We have to speak about hygiene, of what (pregnant women) eat.” “And then, in legal texts, there is the obligation to speak of toxoplasmosis.” “The guidelines dictate 19° in the baby’s room.”

Some participants frequently mentioned their professional experiences: “I worked in a day-nursery and learned that there was a legal text stipulating that nursery furniture was going to be verified because (…) toxic substances could be released.” “In maternity hospitals, when we show them right away that babies’ buttocks need be simply to be washed with soap and water, it is more logical.” A few went so far as to criticize the guidelines: “That’s what’s written down in the recommendations! In actual practice, things are altogether different.” “To deliver 30,000 pieces of advice, it serves no purpose! It is of no interest!” “But are today’s priorities the same as they were 10 or 15 years ago?”

### 3.2. Professionals Attitudes

A picture of a baby’s room was presented to participants in an aim to describe their representations of environmental risks. The gateways to discussion of environmental health were for the most part thematic first in terms of overall environments (nourishment, air pollution, water pollutants), and second of products consumed (products associated with a possible accident in daily life, health care products, nursery equipment) (Table 2). Also, the participants made explicit mention of pollutants (biological, chemical, physical) (Table 3).

The gateways to participants’ references to environmental health were only rarely explicitly related to a child’s health: “*I tell them how important for the child is the room where he stays;*” “*(We need) to anticipate the risks of sudden death;*” “*We know that toxoplasmosis during pregnancy, raises real problems*” or to the mother’s health: “*It is easier (to deal with) when they are overweight.*”

### 3.3. Practices

According to them, health professionals have a role to play in environmental health education. However, if they are legitimate, there are not equipped. To educate population on environmental, matters, environmental health skills and educational skills are required more than medical skills.

The methods listed by members of the focus group to discuss environmental health education are presented in Table 4.

## 4. Discussion

According to them, prenatal professionals have little knowledge in environmental health. A review of the literature showed that out of 21 studies, 13 came to the same conclusion [6], with the exception of considerations on tobacco, whose effects on pregnant women have been widely studied. Recently, a French study came to the same conclusion: Only 17% of health professionals felt able to provide appropriate answers to pregnant women about phthalates [12]. This perception of limited knowledge is in all likelihood associated with the complexity of the field. Indeed, despite numerous in vitro and in vivo (animal and epidemiological) arguments in favor of a relation between exposure to environment pollutants (such as endocrine disruptors) and health, there is no scientific consensus [15]. Absence of consensus could be explained by the variability of exposures associated with a non-monotonous dose–response relationship, diverse molecule mixtures, differing individual susceptibilities, and divergent windows of exposure [16].

In an American study, only 1 out of 15 obstetricians said he had been trained in environmental health, a field in which the participants were nonetheless interested, and would like to be trained [17]. In France, only 17% of the health professional population had heard about environmental health during the course of their initial training [13].

Environmental health training indeed seems highly necessary [18]. It could be carried out during early studies with a dedicated course unit, and during continuous professional training with conferences and regularly updated guidelines taking into account the latest developments in ecology studies; identified reference centers would be involved [6].

In France, some training initiatives have been developed. On a nationwide scale, training in environmental health needs to be included in reference documents pertaining to skills to acquire during medical studies. It would also be desirable that the HAS update its information guidelines on pregnant women.

The prenatal professionals in our study presented an attitude toward families focused on the known risks they had studied in initial training. They perceived environmental health primarily in terms of the biological pollution of environments, a phenomenon having largely to do with environmental hygiene. It should in this respect be mentioned that even though environmental health was diversely evoked and perceived, by definition environmental hygiene refers to the healthiness of the environment, and is by no means the sole constituent of environmental health [19].

During a second phase, it was through acquisition of non-specialized experimental knowledge that prenatal professionals’ perception of environmental health began to focus on emergent risks. Non-specialized knowledge is largely empirical, and it is frequently placed in opposition to scientific knowledge. The two different visions of risk, one holistic and the other analytical, which were categorized by Slovic et al. cited by Chauvin [20] are wholly complementary and equally necessary. They have been theorized in a “psychometric paradigm” of risk perception in which risk is viewed as a combination of science and of psychological, social, cultural, and political factors [21,22,23].

In fact, some authors have shown that experience has a greater impact than scientific knowledge on perception of risk associated with tobacco consumption [24]. It would consequently be desirable in training programs for healthcare professionals that an instructor not only transmit scientific knowledge, but also identify their representations of environmental health.

The perception of environmental health by perinatal professionals through their knowledge of known, emergent, learned, and real-life risks is clearly associated with the social environment and the behavioral characteristics of their patients. Their definition of the environment consequently converges with that of the World Health Organization (WHO), which includes not only the anthropogenic pollutions of environment, but also a number of social and behavioral factors [19]. Their definition of environmental health is likewise concordant with the definition put forward by the WHO in 1994 on the occasion of the declaration of Helsinki: “Environmental health includes the aspects of human health, including the quality of life, which are determined by the physical, chemical, biological, social, psychosocial, and aesthetic factors of our environment [25].

## 5. Conclusions

This study shows that French prenatal professionals working with a socially disadvantaged population perceive environmental health in terms of environmental, largely biological pollution, which is associated with the social environment and a number of behavioral factors. We have shown that in addition to scientific knowledge, non-specialized knowledge is pronouncedly useful. We have recommended the development of training programs and the updating of professional guidelines in view of reducing perinatal environmental exposure, particularly with regard to emergent risks.

## 6. Strengths and Limits of the Study

The study was constructed from a grid allowing for a free and neutral approach, based on representations, without participant responses having exerted any influence. It offered a framework targeting the chosen theme, and was carried out notwithstanding the hierarchical relations among the participants. Indeed, a posteriori evaluation of the training showed that it’s playful, concrete, and situation-based aspect helped to overcome obstacles to discussion of environmental health issues, particularly with regard to the underprivileged. The program also offered an opportunity to consider connections with existing professional practices.

While we organized a single focus group session, during its closure we were attentive to idea saturation, which was favored by the multidisciplinary composition of the group. The target population consisted in prenatal PMI professionals. Given the social characteristics of the populations they were targeting, the results of this study are not generalizable to all health care professionals, and they would need to be confirmed and consolidated in studies involving practitioners working in maternity wards or with their private sector perinatal-centered colleagues.

## Figures and Tables

**Table 1 ijerph-16-02544-t001:** Program of training day.

Timing	Aim	Pedagogic Method
09 h	To welcome	
09:30–11 h	To make emerge representations on environmental health from a non-professional, “citizenly” standpoint	Photolanguage^®^
11–12 h	To sensitize on environmental health and the limits of science in this field	Slideshow
14–16 h	To make emerge representations on environmental health from a non-professional, “citizenly” standpoint	Comments on a Picture of a child’s bedroom

**Table 2 ijerph-16-02544-t002:** Representations of perinatal professionals about environmental risks viewed by thematic.

Thematic with Chronological Order of Presentation	Verbatim	Commentary
Products associated with a possible accident in daily life	*“Because it is right on the ground.”* “*Products that should not be accessible to children, they should be up in a cupboard, either too high to be reached or locked with a key, but in any case, they must not be accessible.”* *“Fire hazards”*	However, the participants did not necessarily know whether or not they were dealing with matters of environmental health: *“I do not know if that object (the car seat, the stroller) is part of the problem.”*
Nourishment	The “*feeding-bottle”* was considered to be a possible source of infectious risk “*We do not give a bottle that has already been used.”* Or else, on the contrary, food preparation was regarded as a means of prevention of infectious diseases: *“As for toxoplasmosis, I tell them to wash vegetables or fruits before eating them…*”	Breastfeeding was also mentioned, as were and the different ways of preparing the bottle: “*What type of water?”* “*Microwaves, bottle warmer, sterilization pan*.” The mother’s diet was likewise alluded to, and one participant stated that “*how the mother is nourished, it is easier (to deal with) when they have diabetes or are overweight.”*
Indoor air pollution:	- Dust: “*The pillow cushion that gathers dust.*” “*I think that this area is full of little dust balls.*” - Cleaning products: “*The pschitt sound when we pass by, which is triggered when you take a breath and let in a bunch of chemicals, it’s awful.*” “*The cleaning products that smell like lily of the valley.*” - Candles: “*lit candles, incense sticks, perfume, essential oil;*” - Toys: “*old toys, old stuffed animals;*” - Do-it-yourself products: “*the can of paint;*” - Bed furnishings: “*the glues and varnishes used in furniture.*”	For the participants, exposure to these pollutants depended on room temperature: “*The warmer it is, the more the products will be released.”*
Healthcare products	*“All those hygiene articles.”* *“It is noticed rather quickly, the mothers who systematically utilize wet wipes; quite often there are rapid repercussions on the babies’ buttocks*.”	
Water pollutants	*“Everything involving the bath, all that is water…(including) water softeners, the number of times it is necessary to give a bath.”*	They were seldom mentioned.
Nursery equipment	For the participants, pregnant women are vulnerable to intense commercial pressure insofar as they are led to believe that “*a properly equipped mother is a good mother;”* this is the case even when their socio-economic circumstances are unfavorable. In some families, “*everything is purchased with the family allowances for the newborn child.”* Among others, purchases are rarer: “*A bed is passed on from one generation to the next*” or involve second-hand goods: “*They cannot buy everything in mint condition.*” The participants proposed simple solutions for purchase of used items, recommending, for example, that when making purchases, glass be preferred to plastic.	

**Table 3 ijerph-16-02544-t003:** Representations of perinatal professionals about environmental risks viewed by pollutants.

Pollutants	Verbatim	Commentary
Biological	*“I focus on infectious risks"* *“It has to do with toxoplasmosis”* *“It has rather to do with germs”* *“That is basic advice”* *“We constantly speak about listeria and toxoplasmosis.”*	Animal allergens, in particular, were stigmatized: “*When there are many animals”* “*I tell myself that it is a very comfortable place for a little cat who wishes to settle down.*” A hygienic vision was central: “*We need to talk about hygiene.*” “*Remember milk’s use-by date.”*
Chemical	More specifically, pregnant women were systematically interrogated on their possible tobacco consumption: “*Ah yes, all patients are asked the question.”*	In addition, bisphenol A was mentioned: “*We do not know if there is bisphenol or not”*, as were pesticides: “*I do not talk about pesticides while monitoring pregnancies”* and solvents, but without any molecule being named: “*any furniture glue,*” “*glues and varnish,*” “*wax,”* “*acrylic paints,”* and “*fixing products.”*
Physical	Electromagnetic fields were “waves”: “*Isn’t it important to know that a particular device uses waves?*” “*Being better connected to the sector, it now functions with waves.*”	Ownership of baby monitoring equipment was another source of preoccupation: *“It is a subject I take up all the time;”* “*a so-called baby phone covers 600 meters* (…), *it crosses tall walls;*” “*it is of no use*.”

**Table 4 ijerph-16-02544-t004:** Methods to discuss environmental health education.

Methods	Verbatim
To take women representations of environmental health into account	*“There is this idea that things which have a good smell means they are healthy.”* *“If it is sold it is that is healthy.”* *”pregnant women have confidence in shops.”*
To discuss risk with solutions of exposure reduction in the same time: Saying the word “risk” is needed in an aim to develop awareness of reality but in a speech with positive health vision	“*I advise to choose this product rather than another.*”
To add value to existing solutions with a targeted speech linked to concrete examples	
To empower women with motivation interviews and with the development of psychosocial skills such as creative opinion	Indeed, women with low social economic status give up to commercial pressure because *“A good mum is a well-equipped mum;” “When I meet a family, I wonder if women are able to ask themselves questions.”*
To proceed step by step, with measurable and realistic objectives	*“If I meet pregnant women for the first time, I won’t speak immediately about environmental risks, I will meet her first and after, according to her, I will try to give prevention messages. Never immediately.*”
To use brochures only if women are able to read them	*“The idea of a brochure depends on which population we have to take care about because some women do not read.*”
To use a simple and visual tool	*“Something demonstrative.”*
To favor collective groups or a minima the presence of the spouse	*“We mixed everybody; it is rare to take a person alone.”*
To favor experiences	*“To show that something is happening.”*

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
