# Peer review of "Environmental Health Knowledge, Attitudes, and Practices of French Prenatal Professionals Working with a Socially Underprivileged Population: A Qualitative Study"

_ijerph, 2019, doi:10.3390/ijerph16142544_

Round 1

Reviewer 1 Report

Albouty-Lialty et al demonstrate the environmental health knowledge, attitudes, and practices of French prenatal professionals working with socially underprivileged population. The content shows originality with the idea and quality content however, there still needs to be a few improvements made to the draft.

1) Please add epidemiological data in introduction. Will make a stronger case going into your methodology and rest of manuscript.

2) If applicable, please write about demographic aspects of participant selection: age, gender, ethnicity, etc.

3) Please do a spell check with correct spelling and grammer (an vs a; then vs than; affect vs. effect; etc).

4) Please make sure words are spaced correctly. See a few words with which seems to be too much spacing in between. 

Author Response

We thank the reviewers for their commentaries. Please find our responses to each point.

0)                 English language and style have been checked again by the English teacher.

1) Please add epidemiological data in introduction. Will make a stronger case going into your methodology and rest of manuscript.

We have added this sentence with an epidemiological reference:

However, no mention is made of exposure to emergent risks such as endocrine disruptors, chemical substances present in numerous sectors and daily objects as plastics, cosmetics, food, even though the medical community has begun to show interest in the problem because of the suspicion about their adverse impacts on the human health [2]. Indeed, as they are likely to affect the major functions of reproduction, growth and development, homeostasis and metabolism, endocrine disruptors are suspected to increase occurrence of cancers, prematurity, fetal growth, obesity or degenerative diseases [3].

2)                 If applicable, please write about demographic aspects of participant selection: age, gender, ethnicity, etc.

We have added this sentence in Methods 2.4, with an epidemiological reference:

As the department head was present, there existed a hierarchical relation between the participants, all of whom were French women. The median age was 42.

3)                 Please do a spell check with correct spelling and grammer (an vs a; then vs than; affect vs. effect; etc).

We did it.

4)                 Please make sure words are spaced correctly. See a few words with which seems to be too much spacing in between. 

We checked all the spaces.

5) In sections 2.3 and 2.8 the authors should provide more details about the photolanguage method, as well as the different sequences of training. Consider to add a scheme or a figure, or both.

We have added this sentence in Methods 2.3 :

The first training sequence, used a photolanguageÒ method. In this method using photos serves as a support for speech so that each one expresses his experiences or representations of a theme. Pictures are placed on a table around which participants can turn. The session begins with a question carefully prepared by the facilitator that leads to the choice of pictures to be silently observed. After the choice, everyone presents his or her photograph whenever he or she wishes, bouncing back or not with the previous speech; others listen carefully without comment. In the training day, the theme chosen was environmental health from a non-professional, “citizenly” standpoint.

6)                 Please write the full name of acronyms HAS and EH.       

We have modified these sentences in Introduction and in discussion sections :

Guidelines and recommendations for pregnant women are given in an official document by the French Authority of Health, la Haute Autorité de Santé (HAS) [1].   

In an American study, only one out of 15 obstetricians said he had been trained in environmental health, a field in which the participants were nonetheless interested, and would like to be trained [17]. In France only 17% of the health professional population had heard about environmental health during the course of their initial training [13].

7)                 In section 3.1, the responses of health professionals were written in italics, then they were not. What is the reason? Did the quoted answers come from the participants?

Yes. We noted that in the article : “All the verbatim coming from the participants are presented in italics”.

8)                 I further suggest the authors to summarize the principal findings of section 3.2 within a table, where the key themes discussed and the main strengths or criticalities emerged may be reported. In this form, reading of results is a bit confusing.

We added a table

Table 2 : Representations of perinatal professionals about environmental risks viewed by thematic

Table 3 : Representations of perinatal professionals about environmental risks viewed by pollutants

As it was clearer with tables, we added two more tables:

Table 1 : Program of training day

Table 4 : Methods to discuss environmental health education

9)                 Reference [20] seems incorrect.

We modified the sentence because reference 21 was about the paradigm of risk perception :

The two different visions of risk, one holistic and the other analytical, which were categorized by Slovic et al. [20] and cited by Lowrance are wholly complementary and equally necessary. They have been theorized in a “psychometric paradigm” of risk perception in which risk is viewed as a combination of science and of psychological, social, cultural and political factors [21,22,23].

Reviewer 2 Report

This manuscript deals with an interesting topic, namely the training of health personnel on the environment and health. Environmental pollutants represent among the main risk factors of health outcomes, and the guidelines of the World Health Organization are aimed at the development of healthy environments as well as at strengthening the resilience of communities residing in contaminated areas. Similarly, 2030 Agenda, the program of United Nations for Sustainable Development, is a plan to protect the planet from degradation, improve quality of life, and protect human health. Since, it is crucial not only to mitigate the impacts of chemical emissions on the environment and activate surveillance systems, but also increase awareness of risks for human health associated to exposure to pollutants, through awareness-raising campaigns and an adequate training of health practitioners, doctors, nurses. Thus, studies like this represent a good starting point to activate specific training programs for healthcare professionals, especially for those working with underprivileged families.

Though the manuscript is generally well written, it requires some improvements.

METHODS

In sections 2.3 and 2.8 the authors should provide more details about the photolanguage method, as well as the different sequences of training. Consider to add a scheme or a figure, or both.

RESULTS

Please write the full name of acronyms HAS and EH.            

In section 3.1, the responses of health professionals were written in italics, then they were not. What is the reason? Did the quoted answers come from the participants?

I further suggest the authors to summarize the principal findings of section 3.2 within a table, where the key themes discussed and the main strengths or criticalities emerged may be reported. In this form, reading of results is a bit confusing.

DISCUSSION

Reference [20] seems incorrect.

Author Response

(The authors gave the same response as above.)

Round 2

Reviewer 2 Report

The manuscript was considerably improved, and the authors addressed my main concerns.

Minor revisions:

In Table 1: please correct  as “11h”

Page 8, line 4, after the colon, please put the lowercase letter.

Page 8, line 16, consider to replace the colon with a dot.

Unfortunately, I still cannot understand why the text refers to Slovic et al. [20] whereas the author of reference [20] is Chauvin.

Author Response

We thank again the reviewer for his commentaries.

Please find enclosed the modifications of the manuscript

1/Table 1: 11h

2/Page 8, line 4 :Recently a French study came to the same conclusion: only 17% of health professionals felt able to provide appropriate answers to pregnant women about phthalates [12]

3/Page 8, line 16 :Environmental health training indeed seems highly necessary [18].  It could be carried out during early studies with a dedicated course unit, and during continuous professional training with conferences and regularly updated guidelines taking into account the latest developments in ecology studies; identified reference centers would be involved [6].

4/Unfortunately, I still cannot understand why the text refers to Slovic et al. [20] whereas the author of reference [20] is Chauvin.

We cited the book of Chauvin who studied the risk perception and the works of Slovic. We modified the sentence by : "The two different visions of risk, one holistic and the other analytical, which were categorized by Slovic et al. cited by Chauvin [20] are wholly complementary and equally necessary. They have been theorized in a “psychometric paradigm” of risk perception in which risk is viewed as a combination of science and of psychological, social, cultural and political factors [21,22,23].
